# The Effect of Intracellular Tacrolimus Exposure on Calcineurin Inhibition in Immediate- and Extended-Release Tacrolimus Formulations

**DOI:** 10.3390/pharmaceutics15051481

**Published:** 2023-05-12

**Authors:** Pere Fontova, Lisanne N. van Merendonk, Anna Vidal-Alabró, Raül Rigo-Bonnin, Gema Cerezo, Stefaan van Oevelen, Oriol Bestard, Edoardo Melilli, Nuria Montero, Ana Coloma, Anna Manonelles, Joan Torras, Josep M. Cruzado, Josep M. Grinyó, Helena Colom, Nuria Lloberas

**Affiliations:** 1Nephrology Department, IDIBELL, Hospital Universitari de Bellvitge, 08907 Barcelona, Spain; perefontova@ub.edu (P.F.); lisannevanmerendonk@hotmail.com (L.N.v.M.); annavidal@ub.edu (A.V.-A.); gcerezo@idibell.cat (G.C.); obestard@vhebron.net (O.B.); emelilli@bellvitgehospital.cat (E.M.); n.montero@bellvitgehospital.cat (N.M.); acoloma@bellvitgehospital.cat (A.C.); amanonelles@bellvitgehospital.cat (A.M.); jtorras@bellvitgehospital.cat (J.T.); jmcruzado@bellvitgehospital.cat (J.M.C.); 2Nephrology Laboratory, Department of Clinical Sciences, Campus Bellvitge, University of Barcelona, 08907 Barcelona, Spain; jgrinyo@ub.edu; 3Biochemistry Department, IDIBELL, Hospital Universitari de Bellvitge, 08907 Barcelona, Spain; raulr@bellvitgehospital.cat; 4Hospital Hartziekenhuis, 2500 Lier, Belgium; stefaan.van.oevelen@heilighartlier.be; 5Biopharmaceutics and Pharmacokinetics Unit, Department of Pharmacy and Pharmaceutical Technology and Physical Chemistry, School of Pharmacy, University of Barcelona, 08028 Barcelona, Spain

**Keywords:** tacrolimus, leukocytes, mononuclear, kidney transplantation, pharmacokinetics, pharmacodynamics

## Abstract

Despite intensive monitoring of whole blood tacrolimus concentrations, acute rejection after kidney transplantation occurs during tacrolimus therapy. Intracellular tacrolimus concentrations could better reflect exposure at the site of action and its pharmacodynamics (PD). Intracellular pharmacokinetic (PK) profile following different tacrolimus formulations (immediate-release (TAC-IR) and extended-release (TAC-LCP)) remains unclear. Therefore, the aim was to study intracellular tacrolimus PK of TAC-IR and TAC-LCP and its correlation with whole blood (WhB) PK and PD. A post-hoc analysis of a prospective, open-label, crossover investigator-driven clinical trial (NCT02961608) was performed. Intracellular and WhB tacrolimus 24 h time-concentration curves were measured in 23 stable kidney transplant recipients. PD analysis was evaluated measuring calcineurin activity (CNA) and simultaneous intracellular PK/PD modelling analysis was conducted. Higher dose-adjusted pre-dose intracellular concentrations (C_0_ and C_24_) and total exposure (AUC_0–24_) values were found for TAC-LCP than TAC-IR. Lower intracellular peak concentration (C_max_) was found after TAC-LCP. Correlations between C_0_, C_24_ and AUC_0–24_ were observed within both formulations. Intracellular kinetics seems to be limited by WhB disposition, in turn, limited by tacrolimus release/absorption processes from both formulations. The faster intracellular elimination after TAC-IR was translated into a more rapid recovery of CNA. An Emax model relating % inhibition and intracellular concentrations, including both formulations, showed an IC_50_, a concentration to achieve 50% CNA inhibition, of 43.9 pg/million cells.

## 1. Introduction

Tacrolimus (TAC) is a calcineurin inhibitor and key immunosuppressant prescribed after kidney transplantation to prevent graft rejection. TAC has a narrow therapeutic window and a large inter- and intra-patient pharmacokinetic (PK) variability. Therefore, therapeutic drug monitoring (TDM) during clinical routine is crucial to decrease the risk of acute rejection and toxicity [1]. Although TDM is currently based on measuring whole blood (WhB) concentrations, the immunosuppressive action is exerted within the lymphocytes and thus, the “in vivo” efficacy should correlate best with intralymphocytic concentrations. While WhB concentrations are influenced by haematocrit levels [2], the free TAC that crosses the biological membranes to reach the intracellular space can be influenced by drug transporters [3,4]. TAC is a substrate of the transporter P-glycoprotein (Pgp, *ABCB1* gene). Genetic polymorphisms of the *ABCB1* gene could determine differences in TAC PKs and pharmacodynamics (PDs) [3,5,6]. Moreover, WhB concentrations can be a poor predictor of intralymphocytic TAC concentrations and of the calcineurin (CN) inhibition profile. In fact, the correlation between pre-dose concentration (C_0_) of WhB and intracellular TAC remains unclear [7,8,9]. Thus, to better reflect the TAC exposure, quantification of TAC directly at the target cell, inside the lymphocytes, has been suggested [10,11]. 

Currently, the area under the curve (AUC) is considered the best measure of systemic exposure, which correlates better with clinical outcomes [12]. However, due to clinical limitations, WhB TAC C_0_ is quantified during TDM since it was corroborated that a WhB C_0_ lower than the therapeutic range increases the risk of acute rejection, while C_0_ above the therapeutic window is correlated with more toxicity. However, acute rejection and toxicity occur even when WhB C_0_ is within the therapeutic range [2,13]. Furthermore, results regarding correlations between WhB C_0_ and AUC are variable among studies [2,14,15,16], possibly indicating that a single C_0_ could be associated with different PK profiles. By contrast, correlation between intracellular TAC C_0_ and rejection severity or lymphocyte’s activation was observed in solid organ transplant recipients [7,17]. 

Knowledge of the relationship between TAC exposure and CN activity is required to monitor TAC therapy, as this PD biomarker directly reflects the biological effect of TAC inside the cells [18,19]. Most PD studies relating TAC exposure with CN activity are based solely on WhB TAC concentrations, with differences in the study design and variables to measure drug exposure and CN activity. A poor or inexistent linear correlation between CN activity and WhB concentrations has been reported [15,20]. In this line, no correlation between CN activity and intracellular TAC concentrations was found in different studies [3,21]. However, these studies were performed in de novo liver transplant recipients shortly after transplantation, and the intracellular PK–PD relationship remains unclear in stable transplant recipients and for once-daily TAC formulations. 

Different administered TAC formulations have variable release/PK profiles and rate/extent of absorption that affects TAC exposure, and this should be considered by clinicians [22]. Only one study explored the impact of different TAC release profiles, specifically the twice-daily immediate-release (TAC-IR) and the once-daily extended-release (TAC-LCP) formulation, on the PK/PD behavior considering WhB concentrations [15]. In this study, the higher peak concentrations after TAC-IR did not result in lower CN activity than TAC-LCP, suggesting a capacity-limited effect [23]. Moreover, a more sustained inhibition was observed after TAC-LCP than with TAC-IR formulation. 

The primary aim of this study was to investigate for the first time how TAC-LCP administration affects intracellular TAC exposure inside peripheral blood mononuclear cells (PBMCs) compared to TAC-IR. Moreover, the PK/PD profiles between intracellular TAC and CN inhibition were also compared with WhB PK/PD profiles after TAC-IR and TAC-LCP in renal transplant patients. 

## 2. Materials and Methods

### 2.1. Study Design 

This was a post-hoc analysis of a prospective, open-label, cross-over, single-center study (NCT02961608) conducted at Bellvitge University Hospital, Spain between 2016 and 2018. Inclusion and exclusion criteria are described in the clinical trial protocol [15]. The study was undertaken in accordance with the Declaration of Helsinki and all participants provided written informed consent. Twenty-five stable kidney transplant recipients administered with TAC-IR (Prograf® Astellas Pharma, Japan or Adoport^®^-Sandoz, Germany) transplanted for at least six months and showing WhB TAC C_0_ between 5 and 10 ng/mL at steady-state were enrolled. After TAC-IR PK/PD sampling, conversion to TAC-LCP (Envarsus^®^-Chiesi Farmaceutici, Italy) was carried out according to the recommended conversion rate of 1:0.7 [TAC-IR:TAC-LCP]. Four weeks post-conversion, a TAC-LCP PK/PD sampling was performed. Blood samples were obtained at pre-dose and at 0.5, 1, 1.5, 2, 3, 4, 6, 8, 12, 12.5, 13, 14, 15, 20 and 24 h post-dosing. All TAC doses were administered in fasting conditions and at least 1 h before meals. 

### 2.2. Measurement of Intracellular and Whole Blood Tacrolimus

Quantification of TAC concentrations in WhB and PBMCs was performed using our previously validated methods [24,25]. Briefly, PBMCs were isolated by Ficoll density gradient and lysed by employing a hypotonic buffer [26]. The extraction of TAC from both PBMC’s lysate and WhB was accomplished by protein precipitation. TAC quantification was performed through ultra-high-performance liquid chromatography coupled with tandem mass-spectrometry method (UHPLC-MS/MS). The WhB and the intracellular TAC concentrations were expressed as ng/mL and pg/million cells, respectively. 

### 2.3. Calcineurin Activity Measurement

Determination of the CN activity in PBMCs was performed using our previously validated method [26]. This method monitored the Ca^2+^ dependent dephosphorylation of a phosphorylated peptide substrate (RIIp). Briefly, reaction buffer and RIIp were added to the lysate of PBMCs. The conversion of RIIp to non-phosphorylated peptide product (RII) by CN occurred, and afterwards, the RII and its corresponding internal-standard (RII-IS) peptides were quantified by UHPLC-MS/MS. CN activity was expressed as pmol RII/min·mg protein. 

### 2.4. Pharmacokinetic and Pharmacodynamic Analysis

#### 2.4.1. Model-Independent Approach

PK parameters from concentration-time profiles at steady-state conditions were determined using the non-compartmental approach. The AUCs from 0 to 24 h (AUC_0–24h_) after TAC-IR and TAC-LCP were calculated by the linear-log trapezoidal rule, peak concentrations (C_max_), time to peak concentrations (T_max_) and trough concentrations before morning intake (C_0_) and at 24 h post-administration (C_24_) and were obtained by direct visual inspection of the data. 

PD parameters were calculated from the CN activity-time profiles. Pre-dose CN activity basal levels at 0 h (I_0_) and 24 h (I_24_), minimum achieved CN inhibition in each patient (I_min_), maximum achieved CN inhibition in each patient (I_nadir_) and time to achieve I_nadir_ (T_nadir_) were estimated from direct visual inspection of profiles. Individual percentages of change in CN activity in regard to I_min_ and I_nadir_ values were calculated according to Equations (1) and (2), respectively.
(1)$${\%\;\mathrm{Inhibition}}_{\mathrm{Imin}}=\left[\frac{\left(I_{\min}-I_{x}\right)}{I_{\min}}\right]\cdot 100$$
(2)$${\%\;\mathrm{Inhibition}}_{\mathrm{Inadir}}=\left[\frac{\left(I_{x}-I_{\mathrm{nadir}}\right)}{I_{\mathrm{nadir}}}\right]\cdot 100$$
where I_x_ was the CN activity at each time-point for each patient. The area under the effect-time profile from 0 to 24 h (AUE_0–24h_) was estimated using the linear trapezoidal rule from these variables. Phoenix WinNonlin program 64 v8.2 (Certara, Princeton, NJ, USA) was used for all PK/PD parameter calculations. 

#### 2.4.2. Pharmacodynamic Modelling

A simultaneous analysis of all CN activity, given by the % inhibition I_nadir_ (Equation (2)) and intracellular concentration profiles, was conducted to estimate the mean PD parameters that best describe the PD relationship. The analysis was performed with a naïve pooled approach using NONMEM v7.4 (ICON Development Solutions, Hanover, MD, USA). The first-order conditional estimation method with interaction was used for PD parameter estimation. The simple (γ = 1) vs. sigmoid inhibitory (γ ≠ 1) E_max_ models with baseline (Equation (3)) vs. without baseline (Equation (4)) were tested to characterize the relationship between intracellular TAC concentrations and responses given by the % inhibition I_nadir_ to remove the influence of different I_nadir_ values amongst patients. The best model selection was guided based on changes in the minimum objective function value (MOFV) and the Akaike information criterion [27].
(3)$$E=E_{0}-\frac{I_{\max}\;\cdot {\;C}^{\gamma }}{{\mathrm{IC}}_{50}{}^{\gamma }+{\;C}^{\gamma }}$$
(4)$$E=E_{0}\cdot \left[1-\frac{C^{\gamma }}{{\mathrm{IC}}_{50}{}^{\gamma }+C^{\gamma }}\right]$$
where E_0_ is the maximum change of % inhibition I_nadir_ observed (Equation (2)), C is the intracellular TAC concentration at each time-point, IC_50_ is the intracellular concentration to achieve 50 % of the maximum change of % inhibition I_nadir_, I_max_ is the maximum change of % inhibition I_nadir_, γ is the sigmoidicity factor. 

Model diagnostics were also based on parameter estimates precision, condition number and goodness-of-fit plots of observed vs. predicted response values. The randomness around the identity line of observed concentrations vs. predictions plots was examined. 

### 2.5. Statistical Analysis 

Baseline demographic characteristics of participants and categorical data were reported as median [interquartile range]. PK and PD parameters were expressed as geometric mean (95% confidence interval), except those considered discrete variables (T_max_, T_nadir_) which were reported as medians (minimum − maximum values). The IC_50_ and E_0_ % CN inhibition were represented as the mean (% relative standard error). Statistical comparison of either PK or PD parameters between formulations was performed using natural log-transformed data for continuous parameters and untransformed data for T_max_ and T_nadir_. A paired *t*-test was applied for continuous data and a Wilcoxon paired test for non-continuous data. Moreover, some PK analyses were adjusted by the total daily dose (TDD). Pearson’s correlation test was applied to analyze correlations between PK and PD parameters. Statistical tests were performed with SPSS Statistics v25 and GraphPad Prism 6.0 software. All tests were 2-sided and considered significant when *p* < 0.05.

## 3. Results

### 3.1. Demographic Data

Twenty-three of twenty-five participants completed the entire intracellular PK study. One participant did not complete the second PK sampling after TAC-LCP administration, and one participant was excluded after conversion due to the commencement of dialysis treatment (exclusion criteria). Baseline demographic and clinical characteristics are shown in Table 1. 

### 3.2. Intracellular Pharmacokinetics

#### 3.2.1. Differences between Intracellular TAC-IR and TAC-LCP Pharmacokinetics

PK profiles and exposure parameters of intracellular TAC reported from 0 to 24 h following administration of TAC-IR and TAC-LCP are shown in Figure 1a and Table 2, respectively. The PK profile normalized by TDD is shown in Figure 1b. The intracellular PK profile of TAC-IR showed rapid absorption after the morning TAC dose, with peak concentrations achieved around 1.5 h, followed by a rapid decrease until 12 h post-administration. The PK profile after the night dose suggested a slower absorption rate with lower peak concentrations and a slower decrease. In contrast, intracellular TAC-LCP profile showed a slower absorption rate compared to TAC-IR, with lower peak concentrations reached around 6 h post-intake (Figure 1a and Table 2). Our results showed that C_0_, C_24_ and AUC_0–24h_ were similar between both formulations (Table 2); however, statistically significant differences were found when normalized by TDD values, observing 30–35% higher values for TAC-LCP than for TAC-IR. Moreover, TAC-LCP showed a 24.1% statistically lower C_max_ compared to TAC-IR; meanwhile, C*_max_*/TDD tended to be higher for TAC-LCP than TAC-IR (Figure 1b and Table 2). Finally, elimination rate (λ*_z_*) was higher for TAC-IR than TAC-LCP, whereas a longer half-life was found for TAC-LCP (Table 2). 

Correlation coefficients between intracellular parameters after TAC-IR and TAC-LCP are presented in Table 2. Significant correlations between trough intracellular concentrations (C_0_ and C_24_) or C_max_ and AUC_0–24h_ were observed in both formulations. Similarly, these high correlations were also obtained when these parameters were normalized by TDD. Moreover, significant correlations with a lower magnitude were seen between trough concentrations and C_max_ in each formulation, also when corrected by TDD. 

#### 3.2.2. Relationship between Intracellular and Whole Blood Tacrolimus

The overlapped mean intracellular and WhB concentration-time profiles of TAC-IR and TAC-LCP are shown in Figure 2A. WhB data were previously reported by Fontova et al. [15]. These plots suggest that no significant delay exists in the time to peak intracellular concentrations (T_max_) compared to WhB TAC after TAC-IR (1.53 [0.83–3.92] and 1.50 [0.68–4.10] h, respectively) [15]. In contrast, Tmax was delayed in intracellular TAC-LCP (6.00 [4.09–7.91] h) compared to data reported in WhB (4.25 [2.98–13.38] h) [15]; however, this delay was not significant (*p* = 0.173) (Figure 2A). Moreover, the C_max_ ratio between both formulations (C_max_ TAC-LCP/TAC-IR) was higher in intracellular (0.78 [0.44–1.92]) than in WhB (0.67 [0.33–1.26], *p* = 0.029), indicating that higher differences between both formulations exist in peak concentrations in WhB than in the intracellular compartment. In this line, when intracellular PK parameters were normalized by WhB parameters, no differences were achieved in C_0_, C_24_ and AUC_0–24h_ between both formulations (Table 3). 

The correlation between intracellular and WhB TAC exposure parameters with or without correction by TDD is shown in Table 3 and Figure 2B. There was a moderate correlation between intracellular and WhB TAC in C_0_ and C_24_ in both formulations, being higher in TAC-LCP formulation. In contrast, the correlation of C_max_ was slightly better in TAC-IR. Correlation in AUC_0–24h_ between WhB and intracellular TAC was only present in TAC-LCP, although in TDD normalized, TAC-IR also reached significance. Comparing all WhB and intracellular TAC concentration values in each formulation (TAC-IR, n = 411; TAC-LCP, n = 343), a significant modest correlation was visible (r ≈ 0.60, *p* < 0.001) (Figure 2B). 

### 3.3. Relationship between Intracellular Tacrolimus and Calcineurin Activity

The CN activity data used in the present study were taken from our previous publication [15]. Figure 2C shows the intracellular TAC and CN activity-time courses after TAC-IR and TAC-LCP. According to these results, a decrease of CN activity from baseline was observed as intracellular TAC concentrations increased for both formulations, similarly to what was observed with WhB in our previous study [15]. Apparently, no delay exists between the maximum CN inhibition and intracellular peak concentrations after both formulations (TAC-IR, 2.00 [0.55–8.02 vs. 1.53 [0.83–3.92]] hours, respectively; TAC-LCP, 5.89 [0.91–12.00] vs. 6.00 [4.09–7.91] hours, respectively). Like the intracellular PK profile, the CN inhibition turns over more rapidly after a morning dose of TAC-IR than after TAC-LCP, although similar turnovers were observed after the night dose (Figure 2C). Moreover, the ratio between AUE_0–24h_ I_min_ (Equation (1)) and intracellular AUC_0–24h_ was higher after TAC-LCP formulation, confirming that higher inhibition was obtained with similar intracellular TAC exposure. 

Inverse correlations between intracellular PK and PD parameters are shown in Table 4. A significant, modest correlation between intracellular C_max_ and I_nadir_ was only observed for TAC-LCP. Similarly, no correlation was observed between the different PD AUEs (AUE_0–24h_ PD I_min_ and AUE_0–24h_ PD I_nadir_) and intracellular C_0_ or AUC_0–24h_, except between AUC_0–24h_ and AUE_0–24h_ PD I_nadir_ for TAC-IR formulation. 

### 3.4. Pharmacodynamic Modelling

As no linear correlation was observed between intracellular TAC and CN inhibition, PD modelling was performed. The model that best described the relationship between intracellular TAC concentrations and the response given by % inhibition I*_nadir_* (Equation (2)) was a simple inhibitory Imax model without baseline (γ = 1) (Equation (4)).

The plots of overlaid observed and predicted % inhibition I*_nadir_* vs. intracellular TAC concentrations suggest that the model adequately described the mean trend of the experimental data (Figure 2D). The plot of observed vs. predicted % inhibition I_nadir_ shows a scattered and random distribution around the identity line (Figure 2E). All parameters were estimated with good precision. The maximum reduction of % inhibition I_nadir_ (E_0_, maximum change of CN inhibition compared to maximum CN inhibition measured within each patient) was 30.4 %, and the IC_50_ value was 43.9 pg/million cells.

## 4. Discussion

This is the first time that intracellular TAC distribution from WhB linked to CN inhibition activity (PD) has been comparatively characterized for TAC-IR and TAC-LCP formulations in kidney transplant patients. PK/PD profiles for 24 h of intracellular TAC concentrations have been compared after TAC-IR and post-conversion to TAC-LCP formulation.

According to our previous works [15,16], TAC distribution kinetics to PBMCs seems to be limited by WhB TAC disposition kinetics and in turn, limited by the TAC release/absorption processes from both formulations. As in WhB [15], a rapid increase of intracellular concentrations were observed after TAC-IR, with a relatively rapid decrease after the TAC peak concentration compared to TAC-LCP showing a more delayed increase and decline of TAC concentrations. The estimated PK parameters support these results. Indeed, the larger T_max_ values after TAC-LCP compared to TAC-IR confirmed its slower intracellular distribution. Whilst similar intracellular C_24_ were found, TAC-LCP showed significantly higher C_24_/TDD than TAC-IR. The slower intracellular distribution from TAC-LCP and the higher bioavailability compared to TAC-IR could explain this result [28]. Likewise, similar intracellular AUC_0–24h_ was achieved after both formulations, but higher AUC_0–24h_/TDD values after TAC-LCP were observed, confirming the higher bioavailability of this formulation [29,30]. The ratio of normalized AUC_0–24h_/TDD values after TAC-IR vs. TAC-LCP administration agreed with the recommended ratio of conversion [1:0.7] in the manufacturers’ labelling for WhB TAC management. As in WhB [15], less fluctuations between peak and trough intracellular concentrations were achieved after TAC-LCP during the dosing interval. The slower TAC release/absorption rate after TAC-LCP contributed to significantly lower intracellular C_max_ values. The lack of significant differences found between C_max_/TDD values was probably due to the lower administered doses for TAC-LCP because of its higher bioavailability. Compared to Francke et al. [28] and Tron et al. [3], similar trough intracellular concentrations after TAC-IR administration were found in our study (24.5 [16–33] vs. 28.4 [9.6–80] vs. 26.8 [9.2–73.5] pg/million cells, respectively), although our administered doses were not the same (5.0 [4.0–7.0] vs. 1.5 [0.5–4.0] vs. 3.0 [2.25–5.0], respectively). The larger range of C_0_ variability reported by Tron et al. [3] and our study compared to Francke et al. [28] could be explained by the lower sample size.

TDM is crucial to confirm existing correlations between trough concentrations and AUC_0–24h_. Previous studies in WhB resulted in correlation values around r = 0.8 for either TAC-IR and TAC-LCP [12,15]. Stronger correlations were found for intracellular values in both formulations, ranging from 0.88 to 0.93 [3]. Unlike WhB, which was only present in TAC-LCP, the intracellular compartment showed a significant correlation between C_0_ and C_max_ in both formulations. Moreover, all these correlations have been corrected for TDD to analyze the magnitude of the dose, showing similar results. All these results supported the use of a single intracellular concentration to reflect total achieved exposure.

This is the first study reporting intracellular TAC distribution after TAC-LCP administration. No influence of the different release/absorption rates on the intracellular/WhB ratios was found. Our C_0_ intracellular/WhB ratios were in line with those reported by Francke et al. [28]. The influence of *ABCB1* polymorphisms on the ratios of C_0_ intracellular/WhB has been recently described [3]. In this line, the low number of patients included limited us to study the influence of *CYP3A5*, *CYP3A4* and *ABCB1* polymorphisms on intracellular concentrations. Controversial results have been reported when analyzing correlations between WhB and intracellular compartments. In our study, the correlation between all concentration values of WhB and intracellular compartments was ρ ≈ 0.60, as previously described [17,28,31,32]. Contrary to some studies [7,8], our results showed a modest correlation between intracellular and WhB C_0_ in TAC-IR, as described by Pensi et al. [31]. In our stable patients, no significant correlation between intracellular and WhB AUC_0–24h_ after TAC-IR was achieved (r = 0.339, *p* = 0.098), although when it was dose-normalized, it reached significance. This relation remains unclear in TAC-IR formulation, since Tron et al. found a correlation (r^2^ < 0.53) within ten days post-transplantation [3], Lemaitre et al. reported a correlation at one day, but not at day seven post-transplantation [21]. On the other hand, TAC-LCP showed a stronger correlation on C_0_, C_24_ and AUC_0–24h_ between intracellular and WhB compartments than TAC-IR. These stronger correlations are advantageous for TAC-LCP, as WhB values seem to better reflect its intracellular exposure. Due to the differences observed in PK profiles and considering the modest correlation observed between intracellular and WhB compartments in both formulations, monitoring WhB concentrations can therefore not be extrapolated to intracellular exposure. These results suggest that other factors influence intracellular exposure, and monitoring TAC in PBMCs could be beneficial to reflect exposure at the site of action.

The PK/PD profiles confirmed that CN inhibition is influenced by intracellular concentrations, with peak concentrations and peak responses occurring without apparent time-delay. Therefore, monitoring peak concentrations can provide helpful information about the maximum inhibitory effect. The faster intracellular elimination from the biophase after TAC-IR administration was translated into a more rapid recovery of CN activity to baseline. By contrast, a slower turnover and a more sustained inhibition was observed after TAC-LCP. Although, largely, no linear correlations between intracellular PK parameters and PD variables were observed in any formulations, they were enhanced in comparison with our previous analysis using WhB parameters [15]. This increased correlation agreed that measuring intracellular TAC would better explain the PD action in the target cells [10,11].

As previously reported [3], the PD relationship between intracellular TAC and the CN inhibition after both formulations was best described by an inhibitory I_max_ model. In our case, the large number of experimental data and the deep steepness observed in CN activity-time profiles allowed a robust estimation of mean IC_50_ values whilst a high variability was found. According to our results, once in the cell, 43.9 pg/million cells of intracellular TAC are required to achieve a % inhibition I_nadir_ equal to the half value of the maximum measured value (E_0_ = 30.4%). The mean estimated E_0_ value (30.4%) suggested that a low reduction of CN inhibition takes place compared with its I_min_ value, probably due to the existing rate-limiting phenomenon, as previously described [15,23]. The study of Tron et al. [3] also described a PD model of TAC-IR formulation but used a different equation in which the I_max_ was modelled and the intracellular C_max_ was incorporated in the model, instead of using all CN activity and intracellular TAC concentration values. In their PD model, similar percentages of maximal inhibition were found (37%), but their IC_50_ values were higher than ones we have obtained (100 vs. 43.9 pg/million cells). This could be explained by the fact that these authors included only peak concentrations in the model; this probably limited the possibility of estimating a more physiologically feasible IC_50_ value. In our model, the observed value of the sigmoidicity factor (n = 1) suggests that one molecule of TAC binds to one of the enzymes. Further modelling analyses with more mechanistic models are required to gain insights into how the different drug release mechanisms can impact the PD profile and turnover times and how it can alter the balance between production and removal of the CN inhibition existing at baseline during a dosing interval. Moreover, these models are best envisaged when the effect measured is a biomarker, such as CN.

The principal limitation was that our study included stable kidney recipients and may not reflect the PK/PD in the first post-transplantation period. Moreover, clinical outcomes analysis was not included in the design of this study. In this context, Francke et al. could not correlate neither WhB nor intracellular TAC with clinical outcomes three months post-transplantation [28]. Further studies should investigate the role of intracellular TAC and CN activity for clinical outcomes.

Patients switching from twice- to once-daily TAC formulations require close PK and clinical monitoring since differences in TAC-IR and TAC-LCP formulations also appear in intracellular PK profiles [12]. This is the first study focusing on intracellular TAC-LCP concentrations, showing a less fluctuating PK profile than TAC-IR, similar to what was observed in WhB [15]. Moreover, intracellular concentrations correlate slightly better to the PD effect of TAC compared to WhB concentrations. Given that the current literature supports the monitoring of intracellular concentrations as a parameter to better reflect TAC exposure, there is room for reconsidering intracellular TAC and/or CN activity monitoring to better explain clinical outcomes.

## Figures and Tables

**Figure 1 pharmaceutics-15-01481-f001:**
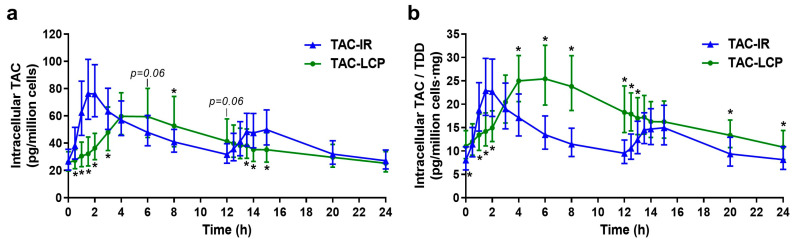
(**a**) Intracellular pharmacokinetic profiles from 0 to 24 h of twice-daily tacrolimus (TAC-IR) and once-daily tacrolimus (TAC-LCP). Intracellular tacrolimus (TAC) concentration inside peripheral blood mononuclear cells (pg/million cells) have been measured. (**b**) Intracellular pharmacokinetic profile adjusted by the total daily dose (TDD). Each point is the geometric mean of all the patients ±95% confidence interval. Paired *t*-test between both formulations was applied. * *p* < 0.05.

**Figure 2 pharmaceutics-15-01481-f002:**
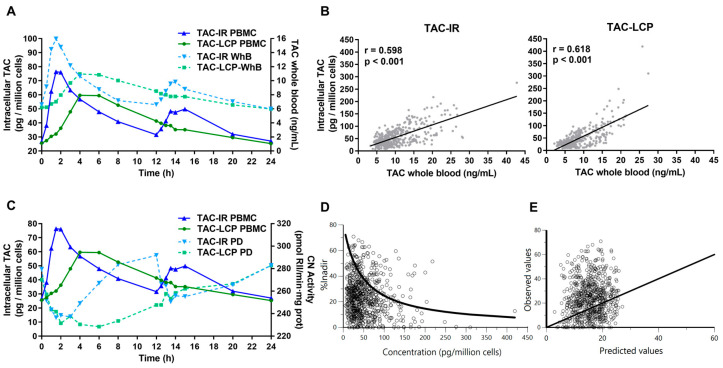
Relationship between intracellular tacrolimus exposure with whole blood pharmacokinetics and its pharmacodynamics. (**A**) Superimposed intracellular and whole blood pharmacokinetic profiles from 0 to 24 h of twice-daily tacrolimus (TAC-IR) and once-daily tacrolimus (TAC-LCP). Blue and green continuous lines represent intracellular tacrolimus (TAC) concentrations (pg/million cells) on the left axis of TAC-IR and TAC-LCP, respectively. Blue and green discontinuous lines show whole blood TAC concentrations (ng/mL) on the right axis of TAC-IR and TAC-LCP, respectively. Each point is the geometric mean of all patients. (**B**) Correlation of intracellular and whole blood TAC concentrations after TAC-IR on the left panel and after TAC-LCP on the right panel. Pearson’s correlation test was used. (**C**) Superimposed pharmacodynamic and intracellular pharmacokinetic profiles from 0 to 24 h of TAC-IR and TAC-LCP. Blue and green continuous lines represent intracellular TAC concentrations (pg/million cells) on the left axis of TAC-IR and TAC-LCP, respectively. Blue and green discontinuous lines represent calcineurin (CN) activity (pmol RII/min·mg prot) on the right axis of TAC-IR and TAC-LCP, respectively. Each point is the geometric mean of all patients. (**D**) Overlaid observed and mean predicted (solid line) % inhibition I_nadir_ values (Equation (2)) vs. intracellular TAC concentrations for the I_max_ pharmacodynamic model. (**E**) Observed vs. predicted % inhibition I_nadir_ values show a scattered and random distribution around the identity line (solid line).

**Table 1 pharmaceutics-15-01481-t001:** Baseline demographic and clinical characteristics.

Characteristics	N = 25
Gender: Male/Female (%)	18/7 (72/28)
Age (years)	58.80 [48.44–71.12]
Body weight (kg)	73.00 [62.75–79.90]
BMI (kg/m^2^)	26.27 [22.53–29.82]
BSA (m^2^)	1.83 [1.68–1.92]
Time post-transplantation (years)	1.84 [0.97–3.88]
Prior kidney transplantation: Yes/No (%)	1/24 (4/96)
Total daily dose (mg)	
- TAC-IR	3.0 [2.25–5.00]
- TAC-LCP	2.0 [1.63–3.50]
Dosing conversion rate [TAC-IR:TAC-LCP]	0.70 [0.67–0.80]
Genotype CYP3A5 polymorphism n (%)	
- **1/*3*	5 (20)
- **3/*3*	20 (80)
Genotype CYP3A4 polymorphism n (%)	
- **1/*1*	24 (96)
- **1/*22*	1 (4)
Genotype ABCB1 polymorphism n (%)	
- **T/*T*	6 (24)
- **C-carriers*	19 (76)

Continuous data are given as median [interquartile range]. Categorical variables are expressed as number and percentage in parenthesis. BMI, body mass index; BSA, body surface area.

**Table 2 pharmaceutics-15-01481-t002:** Exposure parameters of intracellular TAC following administration of TAC-IR and TAC-LCP and correlations between exposure parameters.

**Parameters**	**TAC-IR**		**TAC-LCP**	** *p* **
C_0_ (pg/million cells)	26.78 (9.19–73.45)		25.63 (8.92–79.38)	0.789 ^a^
C_0_/TDD (pg/million cells/mg)	8.03 (1.93–26.71)		10.95 (3.01–38.39)	0.001 ^a^
C_24_ (pg/million cells)	27.09 (6.02–62.23)		25.28 (9.27–95.26)	0.633 ^a^
C_24_/TDD (pg/million cells/mg)	8.13 (1.97–24.34)		10.81 (3.01–40.90)	0.008 ^a^
C_max_ (pg/million cells)	93.06 (31.51–239.41)		70.65 (16.82–256.30)	0.016 ^a^
C_max_/TDD (pg/million cells/mg)	27.37 (11.13–61.54)		30.20 (9.46–92.95)	0.162 ^a^
AUC_0–24h_ (pg/million cells·h)	1052.69 (363.38–2349.86)		969.45 (308.81–3087.34)	0.387 ^a^
AUC_0–24h_/TDD (pg/million cells·h/mg)	315.70 (119.92–878.81)		414.38 (160.13–1141.91)	0.001 ^a^
T_max_ (h)	1.53 [0.83–3.92]		6.00 [3.08–13.92]	<0.001 ^b^
λ_z_	0.0787 (0.0280–0.160) *		0.0461 (0.015–0.110)	<0.001 ^a^
t*_1/2z_* (h)	8.80 (4.33–24.81) *		15.19 (6.34–46.74)	0.001
**Correlations**	**TAC-IR**		**TAC-LCP**	
	**r**	** *p* **	**r**	** *p* **
C_0_ vs. AUC_0–24h_	0.927	<0.001	0.879	<0.001
C_0_/TDD vs. AUC_0–24h_/TDD	0.938	<0.001	0.871	<0.001
C_24_ vs. AUC_0–24h_	0.921	<0.001	0.916	<0.001
C_24_/TDD vs. AUC_0–24h_/TDD	0.945	<0.001	0.921	<0.001
C_max_ vs. AUC_0–24h_	0.898	<0.001	0.961	<0.001
C_max_/TDD vs. AUC_0–24h_/TDD	0.873	<0.001	0.933	<0.001
C_0_ vs. C_max_	0.772	<0.001	0.792	<0.001
C_0_/TDD vs. C_max_/TDD	0.779	<0.001	0.732	<0.001
C_24_ vs. C_max_	0.751	<0.001	0.825	<0.001
C_24_/TDD vs. C_max_/TDD	0.795	<0.001	0.781	<0.001

Data are expressed as geometric mean (95% confidence interval) unless T_max_ is expressed as median [minimum and maximum values]. Pearson’s correlation coefficient (r) was used to analyze the correlation between parameters. C_0_, pre-dose concentration at time 0 h; TDD, total daily dose; C_24_, pre-dose concentration at time 24 h; C_max_, peak concentration; λ_z_, elimination rate constant; t_1/2λz_, elimination half-life; AUC_0–24h_, area under the curve from 0 to 24 h at steady-state; T_max_, time to reach C_max_. * λ_z_ and t_1/2λz_ were estimated from the pharmacokinetic profile of 0–12 h. ^a^ Paired *t*-test; ^b^ Wilcoxon test.

**Table 3 pharmaceutics-15-01481-t003:** Exposure parameters of intracellular tacrolimus corrected by whole blood exposure following administration of TAC-IR and TAC-LCP and correlations between intracellular and whole blood compartments.

**Parameters**	**TAC-IR**		**TAC-LCP**	** *p* **
C_0_ intracellular/C_0_ WhB	4.05 (1.46–9.70)		4.23 (1.70–8.95)	0.595
C_24_ intracellular/C_24_ WhB	4.52 (1.15–11.70)		4.39 (1.20–12.33)	0.942
C_max_ intracellular/C_max_ WhB	5.12 (2.07–9.88)		5.89 (1.32–14.92)	0.041
AUC_0–24h_ intracellular/AUC_0–24h_ WhB	5.03 (1.79–10.02)		4.96 (1.45–10.28)	0.931
**Correlations**	**TAC-IR**		**TAC-LCP**	
	**r**	** *p* **	**r**	** *p* **
C_0_	0.444	0.027	0.583	0.004
C_0/_TDD	0.589	0.002	0.631	0.001
C_24_	0.231	0.265	0.420	0.046
C_24/_TDD	0.585	0.002	0.538	0.008
C_max_	0.686	<0.001	0.567	0.005
C_max/_TDD	0.603	0.001	0.288	0.183
AUC_0–24h_	0.339	0.098	0.572	0.004
AUC_0–24h/_TDD	0.602	0.001	0.423	0.044

Data are expressed as geometric mean (95% confidence interval) unless T_max_ is expressed as median [minimum and maximum values]. Pearson’s correlation coefficient (r) was used to analyze the correlation between parameters. Paired *t*-test was used for comparison between formulations. C_0_, pre-dose concentration at time 0 h; C_24_, pre-dose concentration at time 24 h; C_max_, peak concentration; AUC_0–24h_, area under the curve from 0 to 24 h at steady-state; TDD, total daily dose.

**Table 4 pharmaceutics-15-01481-t004:** Correlations between exposure parameters of intracellular tacrolimus and pharmacodynamic parameters after TAC-IR and TAC-LCP.

Correlations	TAC-IR		TAC-LCP	
	r	*p*	r	*p*
C_0_ vs. I_0_	−0.388	0.055	−0.301	0.153
C_max_ vs. I_nadir_	−0.326	0.112	−0.493	0.014
C_0_ vs. AUE_0–24h_ I_min_	−0.247	0.233	0.212	0.319
AUC_0–24h_ vs. AUE_0–24h_ I_min_	−0.327	0.110	0.236	0.267
C_0_ vs. AUE_0–24h_ I_nadir_	0.325	0.113	0.053	0.806
AUC_0–24h_ vs. AUE_0–24h_ I_nadir_	0.454	0.023	−0.032	0.882

Pearson’s correlation coefficient (r) was used to analyze the correlation between parameters. C_0_, pre-dose concentration at time 0 h; I_0_, calcineurin (CN) activity at time before drug intake (0 h); C_max_, peak concentration; I_nadir_, maximum inhibition of CN activity; AUE_0–24h_ I_min_, area under the activity curve (AUE) from 0 to 24 h of CN inhibition using % inhibition with I_min_ as baseline (Equation (1)); AUC_0–24h_, area under the curve from 0 to 24 h of tacrolimus concentration-time profile; AUE_0–24h_ I_nadir,_ AUE of CN inhibition using % inhibition with I_nadir_ as baseline (Equation (2)).

## Data Availability

Not applicable.

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
