# Peer review of "The Effect of Intracellular Tacrolimus Exposure on Calcineurin Inhibition in Immediate- and Extended-Release Tacrolimus Formulations"

_pharmaceutics, 2023, doi:10.3390/pharmaceutics15051481_

Round 1

Reviewer 1 Report

This article from from Fontova et al. reports intracellular pharmacokinetics and pharmacodynamics of tacrolimus in stable kidney transplant recipients. The originality of this study is the comparison of intracellular PK/PD between IR tacrolimus and LCP-tacrolimus.

The article is well written and the approach is very interesting for the field, especially the comparison between TAC forms.

Some remarks to address before considering publication:

Line 180: Person’s correlation test was applied but with 25 subjects were the variables under a normal distribution? If not the case, another correlation test should be applied such as Spearman. Please justify the use of Pearson test.

Table 1: In total daily dose TAC-IR, it seems that the median value is not reported (only the range). Can it be added?

Table 1: Genotype CYP34A : the frequency of *1/*22 is very high, is there a mix between figures of *1/*1 and *1/*22?

Line 297 and 312: In the text there is references to table 4 and 5 but these tables are missing in the manuscript. So a part of the results are lacking.

Reviewer 2 Report

The paper investigates the intracellular tacrolimus PK of TAC-IR and TAC-LCP and its correlation with whole blood PK and PD. The results provide basic data for the clinical application and drug development of Tacrolimus. Technically I have no other comments.

Some points:

Please unify whether subscripts are required for C0, etc. in the abstract.

Whether “Equation 1”, etc. are marked with brackets as “(1)”.

Reviewer 3 Report

General Comments:

The study shows a more sustained intracellular TAC-LCP concentration than TAC-IR resulting in a more prolonged CN Inhibition which could mean an advantage (regarding clinical point of view). Intracellular concentrations correlate better the PD effect of TAC compared to WhB concentration. The literature supports monitoring of intracellular concentrations to reflect TAC exposure and thus CN activity. A very important conseqence and completion of this finding would be a clinical outcome study, since actually nobody knows if this finding is clinically important. Thus a recommendation should be given to do such a clinical outcome study in 2-3 years, if the drug intake of two different TAC formulations in the here presented cohort has been sustained (that would mean a clinical longtime study). 

Altogether, the present manuscript is a good completion of the precedent publication of the original study of the same working group (Fontova et al. Sustained inhibition of calcineurin activity with a melt-dose once-daily tacrolimus formulation in renal transplant recipients. Clin Pharmacol Ther 2021 Jul;110(1):238-247; Citation No 15 of the present article).

Comments in detail concerning the pharmacokinetic/pharmacodynamic methods:

line 32: it should be shown whether and why pharmacodynamic parameter values for calcineurin activity are more robust from intracellular measurements than from whole blood measurements. 
line 36: the intracellular concentration producing 50 % inhibition of calcineurin activity is 43.9 pg/million cells. This translates into the whole blood IC50 of 9.24 ng/ml [1]. This is less than the IC50 of 15.1 ng/ml  found in volunteers [2] . In stable kidney transplant patients the CE50 concentration was 6.7 ng/ml and comparable [3].
line 136 and 138: the problem with a baseline immunosuppression can only be solved by measuring minimum and maximum caclineurin inhibition in volunteers. such data should be discussed.
line 163: the nadir calcineurin activity is not the maximum effect that might correspond to an even stronger immunosuppression
Table 1: the median for TAC-IR daily dose should be stated ... is it 3.5 mg ... 
figure 1: the intracellular half-life of both tacrolimus preparations should be stated ... for TAC-LCP tha half-life can be read off with 18 hours. 
figure 2 B: by visual inspection, the intracellular IC50 of 43.9 correponds to a whole blood concetration of 7 ng/ml, roughly.
table 3: the correlation between intracellular and whole blood concentration is 4 to 5 and relatively stable. 
line 312: in the available print, the table 5 is completely missing. 
lines 354 and 375: is it really an option for clinical routine to measure intracellular calcineurin activity - if YES, no further cocentration monitoring will be needed 
line 384: the ratio between effect area and concentration area AUEC / AUC would be interesting ... is this ratio different for TAC-LCP and TAC-IR ?
line 396: the Eo = 30.4 % is not the maximum inhibition. this problem can only be solved by a pharmacodynamic study in volunteers. 
line 407: the authors probably mean gamma = 1 not n=1. this means that in their model the Hill coefficient could not be determined or be considered. others report the Hill coefficient with 2.1 and 1.6, respectively [2, 3]  
line 425: the different IC50s are probably an artefact - otherwise, how could it be explained that TAC-IR exhibits a higher IC50 value than TAC-LCP. higher IC50 means less potency. 

Round 2

Reviewer 1 Report

All my remarks have been adressed.